# Bioactive Properties of Breads Made with Sourdough of Hull-Less Barley or Conventional and Pigmented Wheat Flours

Simone Luti [1], Viola Galli [2], Manuel Venturi [2], Lisa Granchi [2], Paolo Paoli [1] and Luigia Pazzagli [1,*]

1 Department of Biomedical Experimental and Clinical Sciences, Università di Firenze, Viale Morgagni 50, 50134 Firenze, Italy; simone.luti@unifi.it (S.L.); paolo.paoli@unifi.it (P.P.)

2 Department of Agriculture, Food, Environment and Forestry, Università di Firenze, Piazzale delle Cascine, 18, 50144 Firenze, Italy; viola.galli@unifi.it (V.G.); manuel.venturi@unifi.it (M.V.); lisa.granchi@unifi.it (L.G.)

* Correspondence: luigia.pazzagli@unifi.it; Tel.: +39-055-2751258

**Featured Application: Exploration of sourdoughs and breads obtained with selected Lactobacilli and flours as source of bioactive peptides for potential applications in functional foods production.**

**Abstract:** Functional and nutritional properties of baked goods can be enhanced by the use of sourdough fermentation, which is defined as a mixture of water and flour fermented by lactobacilli (LAB) and yeasts. Previous data highlighted the ability of sourdoughs obtained with selected LAB strains and commercial flour to produce bioactive peptides equipped with antioxidant and anti-inflammatory properties. More recently, it has been proven that choosing the most suitable combination of selected LAB and high-functional-value flours greatly increase the amount of low-molecular-weight antioxidant compounds responsible for improving the nutritional value of the products. This work aimed to isolate bioactive peptides both from sourdoughs and baked breads obtained with pigmented wheat and barley and a selected combination of LABs. Soluble water extracts were obtained, and low-molecular-weight peptides were isolated. Antioxidant activity was detected by assaying the intracellular ROS production in stressed cultured macrophages, treated with peptides. Moreover, anti-inflammatory activity, highlighted by NFkB pathway inhibition and by COX2 reduction in stressed cells, was demonstrated for peptides obtained from breads. The results allowed the conclusion that the combination of flours and LAB used in the present manuscript led to the production of bakery products with beneficial effects on oxidative and inflammatory status.

**Keywords:** antioxidant activity; anti-inflammatory properties; bioactive peptides; sourdough fermentation

## 1. Introduction

Bread is one of the most widely consumed foods in the world and it is at the base of the Mediterranean diet pyramid. Today, several baked foods are produced by sourdough, a fermented dough that is used worldwide. Sourdough fermentation involves lactic acid bacteria in mixture with yeasts that are responsible for influencing the features of leavened baked goods and may have positive effects on human health [1]. For example, breads obtained by sourdough fermentation show a reduction in the concentration of fermentable oligosaccharides, disaccharides, monosaccharides, and polyols (FODMAPs) whose intake is associated with the onset of irritable bowel syndrome symptoms. FODMAPS present differences in carbohydrate digestibility and absorption and a lower post-prandial glycaemia response [2]. Moreover, a better digestibility of sourdough compared to baker's yeast breads was recently observed [3], as well as the ability to decrease the release of pro-inflammatory cytokines induced by amylase/trypsin inhibitors and that are degraded during the fermentation [4].

All this set of healthy effects can be due to the type of grain flours, on the one hand, and to the mixture of lactobacilli (LAB) and yeasts used for fermentation, on the other [5]. The metabolism of sourdough microbiota, the activity of cereal enzymes, and the peculiar composition of the grains work together to determine the presence of low-molecular-weight compounds, phenols, and peptides responsible for improving the nutritional value of the baked product [6,7].

The formation of a stable microbiota is difficult to standardize, and the use of microbial starters able to guarantee fast, efficient, controllable, and reproducible fermentations may be the method of choice for obtaining baked goods equipped with technological, sensory, and nutritional properties [5]. LAB selection consists in the study of a large number of strains that are characterized for various activities, growth and acidification rate, and capability to increase or to produce bioactive compounds, such as peptides and amino acid derivatives [5,8]. The screening of numerous isolates of LAB from Italian sourdoughs obtained with commercial wheat flour (type "00") led to the identification of three strains belonging to *Companilactobacillus farciminis* (formerly known as *Lactobacillus farciminis*, [9]), *Furfurilactobacillus rossiae* (formerly known as *Lactobacillus rossiae*), and *Fructilactobacillus sanfranciscensis* (formerly known as *Lactobacillus sanfranciscensis*), which were selected for their ability to produce bioactive peptides endowed with antioxidant and anti-inflammatory activity [8]. Bioactive peptides also retained their functional properties in the breads prepared with the same sourdoughs [10]. Depending on the starting material sourdoughs or breads, the peptides have different amino acid compositions, which in any case does not imply changes in their biological activity [10].

On the other hand, the final features of sourdoughs and breads are closely connected to the type of grain used during the baking process. Whole grain and particularly pigmented cereals are an important source of bioactive compounds, micronutrients, and dietary fiber [11–13]. In particular, some pigmented whole grains are innovative and valuable raw materials for the production of functional foods. Together with phenolic acids, these grains show significant levels of anthocyanins and carotenoids, bioactive compounds responsible for their characteristic blue-purple and yellow-orange colors, and for their nutritional value [14,15]. The nutritional quality of breads can also be related to the content of beta-glucans, of which hull-less barley is rich in; consequently, it has been recently proposed for the manufacture of functional foods [16].

Therefore, determining the most suitable combination of selected LAB and high-functional-value flour can be the way to produce healthy baked goods. A recent study, dealing with the selection of lactobacilli strains, used sourdoughs prepared with a blue-grain wheat flour (cv Skorpion), a conventional red-grained wheat variety (cv Aubusson), and a hull-less barley flour (cv Rondo), which allowed the identification of some lactic acid bacteria strains, best equipped with proteolytic and peptidase activities, optimal pro technological properties, and antioxidant activity [5]. In this manuscript, the selected LAB strains were used in combination with the flours and added along with *S. cerevisiae* strains in order to obtain sourdoughs and breads with putative nutraceutical properties. In particular, the bioactive peptides produced during fermentation were isolated from aqueous extracts fractionated by chromatography and used in ex vivo assays to highlight the ability both to reduce intracellular reactive oxygen species production and to decrease the expression of some inflammatory markers.

## 2. Materials and Methods

### 2.1. The Flours and Ingredients

Three different whole grain flours were used for the sourdough preparation. Two varieties of bread-making winter common wheat (*Triticum aestivum* spp. *aestivum* L.): a conventional red-grained variety (cv Aubusson, Limagrain Italia SpA) and a blue-grained variety rich in anthocyanins (cv Skorpion, Agricultural Research Institute Kromeriz, Ltd., Kromeriz, Czech Republic). In addition, a hull-less spring barley variety (*Hordeum vulgare* L. var. *nudum Hook*, cv Rondo, Società Italiana Sementi s.p.a., San Lazzaro di Savena, Italy)

was considered. Type "00" organic wheat flour (Coop, Casalecchio di Reno, Bologna, Italy) was used for bread making, and breads were also fortified with fermented brans, provided by University of Bari and prepared according to [16].

### 2.2. Sourdough Preparation

Flours were used to prepare three sourdough with different combinations of lactic acid bacteria and yeasts, previously characterized for their functional capabilities [5,17,18]. The sourdough prepared with only wheat flour cv Aubusson was inoculated with *Lactiplantibacillus plantarum* Fi13, *Levilactobacillus brevis* LD66, and *Saccharomyces cerevisiae* D20Y; the sourdough prepared with only wheat flour cv Skorpion was inoculated with *L. plantarum* Fi58, *Fructilactobacillus sanfranciscensis* Fi33, and *S. cerevisiae* L6Y; and the dough yield ((amount of flour + amount of water) × 100/(amount of flour)) was 180. Sourdough prepared with only hull-less barley cv Rondo was inoculated with *L. plantarum* Fi13, *Fr. sanfranciscensis* Fi33, and *S. cerevisiae* L6Y; and the dough yield was 200. Microorganisms were grown for 24 h at 30 °C, LAB in MR3i medium [19], yeasts in YM medium, then recovered by centrifugation (10,000× *g* for 10 min), washed with sterile physiological solution, and resuspended in distilled water to inoculate ca 7 log CFU/g and ca 6 log CFU/g, respectively. Sourdoughs were fermented for 12 h at 30 °C before being added as an ingredient for bread-making.

### 2.3. Bread-Making

For bread production, doughs were prepared according to the recipes reported in Table 1. Sourdough was added as 10% (*w/w*), Type "00" organic wheat flour was used, and breads were fortified with fermented brans.

**Table 1.** Ingredients for bread-making. Aubusson: bread with a sourdough made with wheat flour cv Aubusson; Skorpion: dough with a sourdough made with wheat flour cv Skorpion; Rondo: dough with a sourdough made with barley flour cv Rondo.

| | Bread | | |
|---|---|---|---|
| Ingredients | Aubusson | Skorpion | Rondo |
| Sourdough (g) | 200 | 200 | 200 |
| Wheat flour Type 00 (g) | 985 | 985 | 1010 |
| Fermented bran (g) | 40 | 40 | 40 |
| Water (mL) | 815 | 815 | 790 |
| Dough yield | 180 | 180 | 180 |

All the ingredients were added at the same time and mixed for 10 min in a twin-arm mixer (RS12, Bernardi, Villar San Costanzo, Cuneo, Italy). After that, the doughs were divided into molds of 100 g, placed in the trays in a proofing chamber at 30 °C with 88–90% relative humidity for 4 h. Samples were baked for 15 min at 200 °C in an oven (Rossella, Unox, Padua, Italy), with vapor injection in the first instants of baking.

### 2.4. Microbiological Analysis and Determination of pH and Total Titratable Acidity (TTA)

Ten grams of dough sample were transferred into 90 mL of sterile physiological solution and homogenized for 2 min in a Stomacher Lab Blender 400 (Seward Ltd., Worthing, West Sussex, UK). After decimal dilutions, 100 µL of these suspensions were plated for cell enumeration using MR3i medium for the LAB [19] and MYPG for the yeasts, using the pour plate method. LABs were counted after incubation for 48–72 h at 30 °C under anaerobic conditions. Yeasts, plated on MYPG agar containing sodium propionate (2 g/L), were counted after incubation for 48 h at 30 °C under aerobic conditions. Plate counts were performed in duplicate for each dilution. The pH values were determined by a pH-meter (Metrohm Italiana Srl, Varese, Italy). The TTA was measured on 10 g of dough sample,

which were homogenized with 90 mL of distilled water for 3 min and expressed as the amount (mL) of 0.1N NaOH to achieve a pH of 8.5.

### 2.5. Extraction and Fractionation of Water-Soluble Extracts (WSEs)

The water-soluble extracts (WSEs) were obtained by extracting dried breads and sourdoughs at the end of fermentation, with sterile water (1:3 *w/v*), and then centrifuged at 14,000× *g* for 20 min at 4 °C. Soluble peptides were obtained from the WSEs by liquid chromatography according to the method previously used by Luti et al. [10]. Briefly, water liquid extracts were assayed for protein and peptide content by the BCA method (BicinChoninic Acid, Pierce Chemical, Rockford, IL, USA). In total, 5 mg/mL ca of each sample was added to 0.05% (*v/v*) trifluoroacetic acid (TFA) and centrifuged at 10,000× *g* for 10 min to remove impurities. Samples were analyzed by the C18-reverse phase HPLC (Thermo Fisher Scientific, MA, USA) on a Vydac column (4.6 × 250 mm; 5 μm; 300 Å, Columbia, MD, USA). Elution was performed using a water/acetonitrile gradient in the presence of 10 mM trifluoroacetic acid (TFA). The eluted peptides were fractionated according to the retention times as low molecular weight (LMW: 5–20 min). Collected fractions were lyophilized, re-dissolved in bi-distilled water, and assayed for their peptide content to obtain a final concentration of 0.5 mg/mL.

### 2.6. Cell Cultures for Ex Vivo Assays

Murine macrophages (RAW 264.7, Sigma Aldrich, St. Louis, MO, USA) were cultured in standard conditions as previously reported [10]. The cells were routinely sub-cultured every two days and used for biological activity determination of isolated LMW peptide fractions.

#### 2.6.1. Cell Toxicity Assays

The cytotoxic activity of LMW peptides was evaluated using the MTT method. This assay is based on the conversion of MTT (3-4,5-dimethylthiazol-2-yl-2,5-diphenyltetrazolium bromide, Sigma-Aldrich, St. Louis, MO, USA) into formazan crystals by living cells and it was used to determine the percentage of cell death upon treatment with LMW peptides from sourdough [20]. Cells were plated on 24-well plates in 500 μL of fresh medium at a density of $2 \times 10^5$ cells per well. After 24 h, aliquots of the LMW extracts obtained from sourdoughs were added at a final concentration of 0.05 mg/mL and incubated at 37 °C for a further 24 h. A negative control was obtained using cells without the addition of LMW-extracts; cells treated with 1 μg/mL lipopolysaccharide (LPS) were used as the positive control. After incubation, 250 μL of 0.5 mg/mL MTT in DMEM (Dulbecco's Modified Eagle's medium) without red phenol were added and incubated in the dark at 37 °C for 45 min. Then, 200 μL of DMSO (dimethylsulfoxide) were added and the absorbance was quantified at 595 nm with a microplate reader (BioRad, Hercules, CA, USA). A lower absorbance value indicates a higher cell death.

#### 2.6.2. Recovery of Cell Viability

LMW peptides obtained from breads were used to treat $2 \times 10^5$ RAW 264.7 cells in a 24-well plate according to the procedure reported and then they were assayed with Crystal Violet (Sigma-Aldrich, St. Louis, MO, USA) to detect cell viability [21]. Briefly, at the end of the treatment, cells were washed with 0.5 mL of phosphate-buffered saline (PBS, Sigma Aldrich) to remove the dead cells and 200 μL of 1% Crystal Violet (*w/v*) were added for 15 min under shaking. Subsequently, Crystal Violet solution was removed, and cells were washed with PBS several times to eliminate the unreacted dye. Finally, cells were lysed with 2% SDS (sodium dodecyl sulfate) and the absorbance at 595 nm was read with a microplate reader.

### 2.6.3. Intracellular Reactive Oxygen Species (ROS) Measurement

The assessment of intracellular ROS was done using RAW 264.7 cells according to the method reported by Galli et al. [8]. Experiments were performed on 24-well multiplate at a cell density of $5 \times 10^5$ per well. Cells were treated with LMW peptides at the final concentration of 0.05 mg/mL for 1 h and then co-incubated with 1 µg/mL of LPS for 24 h. The pretreatment with peptides and the concentration used were chosen on the basis of previous works and on the literature data [8,10,22]. Stressed cells were used as the positive control, and cells treated with 0.05 mM ascorbic acid as the negative control. After incubation, 10 µM of $H_2$DCF-DA (2′,7′-dichlorodihydrofluorescein diacetate) was added for 30 min in the dark and then the cells were lysed with the RIPA buffer (50 mM TRIS-HCl, 150 mM NaCl, 100 mM NaF, 2 mM EGTA, 1% Triton X-100) and centrifuged at $10,000 \times g$ for 10 min. The fluorescence was measured using a microplate reader at 485 and 538 nm excitation and emission, respectively (Fluoroscan Ascent FL, Thermo Electronic Corporation, MA, USA). The values were determined as fluorescence intensity units and the results expressed as a percentage of the reduction in ROS formation.

### 2.6.4. Immunoblot Analysis

RAW 264.7 cells ($5 \times 10^5$) were plated in a 24-well multiplate and treated with LMW peptides as reported above. Controls were also prepared treating cells with LPS alone (negative control) and with LPS plus 2 mM acetyl salicylic acid (positive control), respectively. After 24 h of incubation, the cells were treated according to current literature methods to determine the levels of some anti-inflammatory markers [10,22]. Cells were lysed in the Laemmli buffer and 50 µg of each sample were resolved by 10% SDS-PAGE and then transferred onto PVDF (polyvinylidene fluoride) membrane. The blotted membranes were saturated with 5% bovine serum albumin (BSA) and 0.1% Tween 20. They were kept in incubation for 1 h at room temperature (25 °C) and finally incubated overnight with the primary antibodies diluted to 1:1000. Anti-pNFkB 93H1, NFkB D14E12, and IkB 44D4 (Cell Signaling Technology, Inc. Danvers, MA, USA) were used as markers of the nuclear factor NF-kB pathway [23,24]. Anti-COX2 C-20 (Santa Crux Biotechnology, Dallas, TX, USA) antibodies were used to detect the expression levels of cyclooxygenase 2 [25,26]. Anti-actin13E5 antibodies (Cell Signaling) were used to detect actin, as a reference protein.

After further washing steps, the membranes were incubated with horseradish peroxidase-conjugated secondary anti-rabbit antibodies (Cell Signaling) and diluted at 1:2000 in PBS containing 5% BSA and 0.1% Tween 20 for 1 h. An ECL kit (enhanced chemiluminescence, GE healthcare) was used to obtain chemiluminescence signals that were acquired with a molecular imaging station from Kodak. Values obtained from the densitometric analysis were normalized on the actin signal.

### 2.7. Statistical Analysis

The numerical results of microbial and chemical analysis in this study are the averages of three independent replicates. Data were analyzed by one-way analysis of variance (ANOVA). The means comparisons were determined by Tukey's test ($p < 0.05$). The results of ex vivo assays are presented as the mean values $\pm$ SD of three separate experiments for each test. The level of statistical significance was determined using Student's *t*-test (for comparisons between two groups). A *p* value of <0.05 was considered to be significant. GraphPAD Prism6 Software, (San Diego, CA, USA) was used.

## 3. Results

### 3.1. Dough Analysis

The three sourdoughs decreased their pH after 12 h of fermentation, reaching values of $4.08 \pm 0.02$ in the Aubusson sample, $4.24 \pm 0.00$ in the Skorpion sample, and $4.19 \pm 0.06$ in the Rondo sample. The LAB and yeast final concentrations did not point out any differences among the sourdough: LAB was ca 9.3 log CFU/g and yeast was 7.0 log CFU/g.

The results of the acidification, volume increase, and cell concentrations of the doughs before baking bread are reported in Table 2.

**Table 2.** Acidification (pH, total titratable acidity), volume increase (%, $\Delta V/V0$), and microorganism concentrations (LAB and yeast) of the doughs at the end of the 4-h fermentation. $\Delta$: difference between the final and the initial value. Aubusson: dough with a sourdough made with wheat flour cv Aubusson; Rondo: dough with a sourdough made with barley flour cv Rondo. Results are expressed as average $\pm$ standard deviation.

| Dough | Final pH | $\Delta$pH | Final TTA | $\Delta V/V0 \times 100$ | LAB (CFU/g) | Yeasts (CFU/g) |
|---|---|---|---|---|---|---|
| Aubusson | $4.37 \pm 0.20$ | $0.64 \pm 0.11$ | $5.10 \pm 0.6$ | $67 \pm 5$ [b] | $(8.75 \pm 0.83) \times 10^8$ | $(5.83 \pm 0.73) \times 10^6$ |
| Skorpion | $4.39 \pm 0.25$ | $0.70 \pm 0.11$ | $5.00 \pm 0.1$ | $94 \pm 8$ [c] | $(8.15 \pm 0.92) \times 10^8$ | $(7.55 \pm 0.82) \times 10^6$ |
| Rondo | $4.42 \pm 0.16$ | $0.67 \pm 0.04$ | $4.35 \pm 0.5$ | $11 \pm 0$ [a] | $(8.65 \pm 1.34) \times 10^8$ | $(6.45 \pm 0.64) \times 10^6$ |

Values in the same column with different letters ([a–c]) are significantly different ($p < 0.05$).

The doughs decreased their pH by more than 0.60 unit in 4 h, with the final pH ranging from 4.3–4.4, which was not statistically different among the samples. The Skorpion sample showed the highest volume increase of the sourdough bread. Rondo bread barely increased in volume due to the poor technological quality of barley flour. As expected, the control bread leavened with baker's yeast reached the highest final volume. No differences were found regarding microorganism concentrations, for both lactic acid bacteria and yeasts in sourdough bread that were in the typical range of sourdough products [19].

*3.2. Low-Molecular-Weight Peptides Fractionation*

WSEs obtained from sourdoughs and breads prepared with the different combination of microorganisms and flours were freeze-dried, re-suspended in water with 0.1% TFA added, and finally, 500-μL aliquots were applied on reverse-phase chromatography (Figure 1). A separation between low and high molecular species is clearly visible in the chromatographic layouts, thus enabling us to collect fractions by hand-picking: the LMW peptides were recovered either for doughs (Figure 1a) and breads (Figure 1b) based on the retention times.

For each combination, the 5–20-min fractions were collected and analyzed by SDS-PAGE (Figure 1c). On the basis of their retention times and the protein content, the fractions were called LMW peptides, assayed to determine the protein content, lyophilized, and finally re-suspended in bi-distilled water at the same protein concentration (0.05 mg/mL) useful to treat cell cultures according to Table 3.

**Table 3.** LMW peptides from doughs and breads: flours, microorganism used for fermentation, and peptide concentration. Protein was determined by the BCA method and expressed in mg/mL.

| | Flour | Bacteria and Yeasts | mg/mL |
|---|---|---|---|
| Doughs | Aubusson | *L. plantarum* Fi13<br>*L. brevis* LD66<br>*S. cerevisiae* D20Y | 4.44 |
| | Skorpion | *L. plantarum* Fi58<br>*L. sanfranciscensis* Fi33<br>*S. cerevisiae* L6Y | 5.17 |
| | Rondo | *L. plantarum* Fi13<br>*L. sanfranciscensis* Fi33<br>*S. cerevisiae* L6Y | 3.76 |

**Table 3.** *Cont.*

|  | **Flour** | **Bacteria and Yeasts** | **mg/mL** |
|---|---|---|---|
| Breads | Aubusson | *L. plantarum* Fi13<br>*L. brevis* LD66<br>*S. cerevisiae* D20Y | 7.01 |
|  | Skorpion | *L. plantarum* Fi58<br>*L. sanfranciscensis* Fi33<br>*S. cerevisiae* L6Y | 5.39 |
|  | Rondo | *L. plantarum* Fi13<br>*L. sanfranciscensis* Fi33<br>*S. cerevisiae* L6Y | 5.43 |

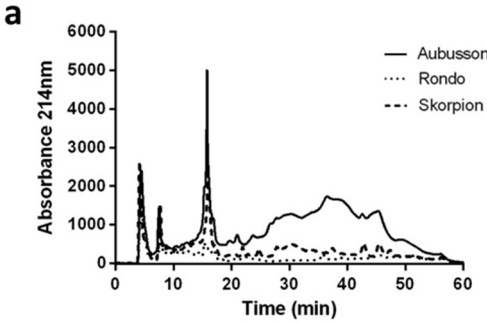

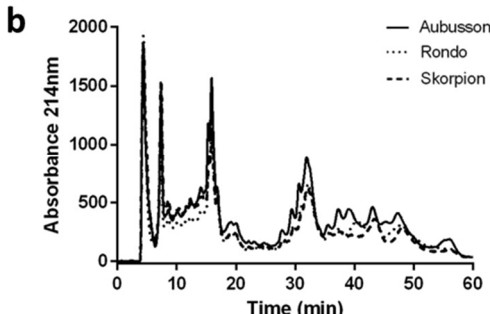

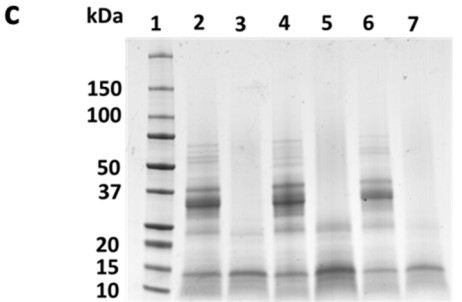

**Figure 1.** HPLC fractionation of WSEs from doughs (**a**) and breads (**b**). First, 500- μL aliquots of each sample (Aubusson, Skorpion Rondo) were applied on a reverse phase column (Vydac, 4.6 × 250 mm; 5 μm; 300 Å) and eluted with a non-linear gradient (Solvent A: $H_2O$ + 10 mM TFA; Solvent B: $CH_3CN$ + 10 mM TFA. Gradient: 0–10 min 30%B, 10–50 min 50%B, 50–60 min 100%B). (**c**) SDS-PAGE analysis of extracts and peptides from breads. 1 Bio-rad All Blue Standard; 2 Aubusson; 3 Aubusson 5–20 min; 4 Rondo; 5 Rondo 5–20 min; 6 Skorpion; 7 Skorpion 5–20 min.

### 3.3. Cytotoxicity Assay

The MTT test was used to assess the lack of cytotoxicity of peptides obtained from both doughs (Figure 2a) and breads (Figure 2b) on cultured murine macrophages. Cells were treated with LMW peptides for 24 h; the positive control was obtained using LPS to reduce cell viability as a consequence of the activation of inflammatory pathways and induction of the synthesis of reactive oxygen species [27].

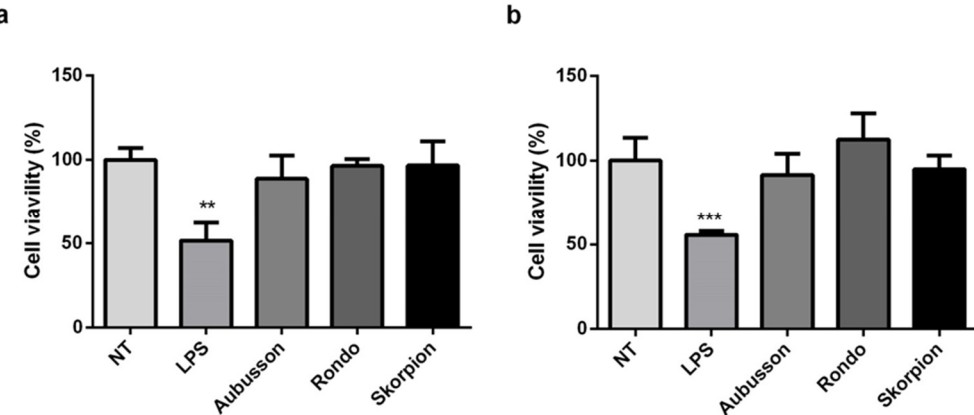

**Figure 2.** Effects of LMW peptides from doughs (**a**) and breads (**b**) on cell viability. RAW 264.7 were incubated with LMW peptides (0.05 mg/mL) or LPS for 24 h. After this time, cell viability was evaluated using the MTT test. The absorbance of formazan was determined at 595 nm. NT: negative control, untreated cells; LPS: positive control, cells treated with the only LPS; Aubusson, Rondo, and Skorpion indicate cells treated with LMW peptides obtained from doughs and breads prepared with the respective flours. Data reported in the figure represent the mean ($\pm$SD) of three different experiments performed in duplicate. Statistical analysis was performed for each sample vs. LPS (** $p < 0.01$; *** $p < 0.001$).

Figure 2 reports the percentage of viable cells recovered after treatments and clearly shows that none of the mixture of peptides induced a significant reduction of cell viability if compared with the respective negative control (NT, non-treated cells). LMW peptides from breads obtained with Rondo (Figure 2b) even seemed to have a positive effect on cell viability; however, this is not significant if compared to the value obtained for the negative control. The values from LPS-treated cells indicate that the method is suitable to detect the putative effect of stressful molecules on cell viability as widely reported in the literature [20,27,28].

### 3.4. Antioxidant Activity of LMW Peptides

We tested the ability of the LMW peptides, both from sourdoughs and breads, to inhibit endogenous cellular ROS production by co-incubating RAW cells with LPS and peptides. Consequently, the effect of LMW peptides was measured as the ability to reverse the ROS production induced by LPS itself. Figure 3 shows that each sample, both from sourdoughs (Figure 3a) and breads (Figure 3b), is able to statistically decrease the amount of ROS that formed in cells upon LPS treatment. Cells were also treated with ascorbic acid, a well-known antioxidant molecule used as a reference to verify the goodness of the method.

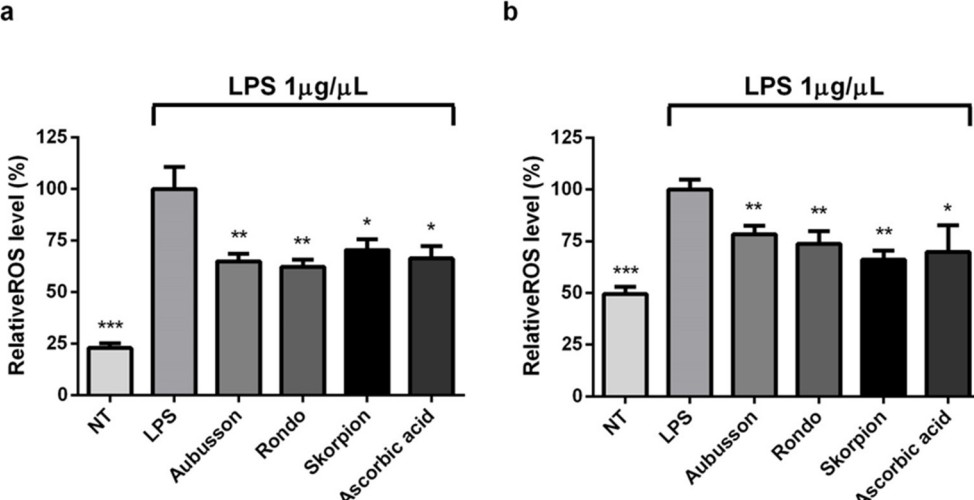

**Figure 3.** Effect of LMW peptides from doughs (**a**) and breads (**b**) on intracellular ROS. Cells were treated with 0.05 mg/mL LMW fractions for 1 h and then co-incubated with 1 μg/mL LPS for 24 h as reported in the material and methods section. NT: negative control, untreated cells; LPS: positive control, cells treated with only LPS; AA: cell treated with 0.05 mM ascorbic acid. Results are the mean (±SD) of three different experiments performed in duplicate. Statistical analysis was performed for each sample vs. the LPS value (* $p < 0.05$; ** $p < 0.01$; *** $p < 0.001$).

All the peptides obtained both from sourdoughs and breads were able to reduce the oxidative status of the cell as much as the 0.05 mM ascorbic acid does. Despite the different values of ROS levels obtained, the reduction of the ROS level was about 25–30% if compared with the respective LPS value. Peptides from Skorpion flour appear to have a greater antioxidant effect when obtained from bread, while peptides from Aubusson and Rondo seem to have a stronger antioxidant effect when prepared from dough, but, however, the differences among peptides from different samples are not significant if compared to each other.

### 3.5. Recovery of Cell Vitality and Anti-Inflammatory Activity

The role of LMW peptides from breads was further detected by assaying both cell vitality and anti-inflammatory activity on cultured cells stressed with LPS, and the crystal violet method was used to stain live cells that remained attached to culture plates while dead cells were detached and washed away [29]. The percentage of viable cells in the sample treated with LMW peptides and LPS clearly indicates that the presence of peptides has a protective effect on the LPS-induced stress (Figure 4). LMW peptides from breads obtained from the Aubusson breads show same vitality of the negative control, to indicate that peptides reverse the negative effect of LPS. In addition, the peptides from both Rondo and Skorpion breads even increase the number of viable cells, playing a healthy role regarding the LPS-induced effects.

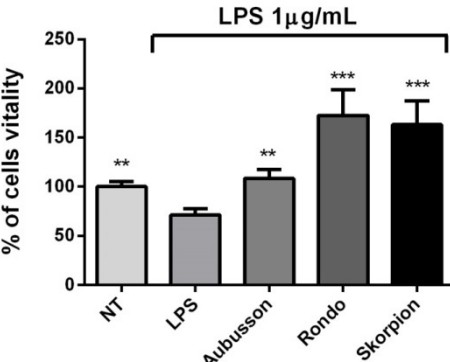

**Figure 4.** Recovery of cell viability. RAW 264.7 cells were incubated with LMW peptides (0.05 mg/mL) from the breads for 1 h and then with LPS for 24 h. Cell viability was evaluated using the crystal violet test. The absorbance due to viable cells was read at 595 nm. NT: negative control, untreated cells; LPS: positive control, cells treated with the only LPS; Aubusson, Rondo, and Skorpion indicate cells treated with both LPS and LMW peptides. Data reported in the figure represent the mean ($\pm$SD) of three different experiments performed in duplicate. Statistical analysis was performed for each sample vs. LPS. (\*\* $p < 0.01$; \*\*\* $p < 0.001$).

The same treatment scheme was also used to unravel the ability of peptides from breads to reverse the LPS-induced inflammation. Upstream markers, either as the ratio between the phosphorylated and total form (pNFkB/NFkB) of the transcription factor NFkB (nuclear factor-kappa B) [23], and the expression of its inhibitor, the IkB factor, were determined for peptides from breads. Moreover, COX2 (cyclooxygenase 2), which remains unexpressed under normal conditions and increases during inflammation, was chosen as a downstream marker of the inflammatory response [25,26]. Immunoblot analysis was performed using antibodies to detect the expression levels of these factors belonging to the inflammatory pathway. Figure 5 shows the results from blots obtained after cell lysis and detection with the specific antibody.

The pNFkB/NFkB ratio (Figure 5a) significantly decreased in macrophages treated with LMW peptides and the percentage of reduction is comparable with that obtained by acetyl salicylic acid treatment and is in agreement with the observation that many anti-inflammatory molecules exert their action by inhibiting nuclear translocation of NFkB, which occurs upon phosphorylation [30]. Results from the IkB detection (Figure 5b) show the same trend: the increase in the IkB value goes together with the decrease in NFkB translocation, as expected for anti-inflammatory compounds. IkB expression values obtained from cells treated with breads are lower than values detected upon ASA treatment; nevertheless, peptides from Aubusson, Rondo, and Skorpion breads are able to increase, in a significant manner, the IkB expression when compared to IkB expression determined for the positive control. Finally, COX2 expression, which is not present in the negative control, is reduced in a significant manner only by peptides from Rondo and Skorpion breads. On the contrary, peptides from the Aubusson sample slightly reduce the expression of COX2, but not significantly (Figure 5c).

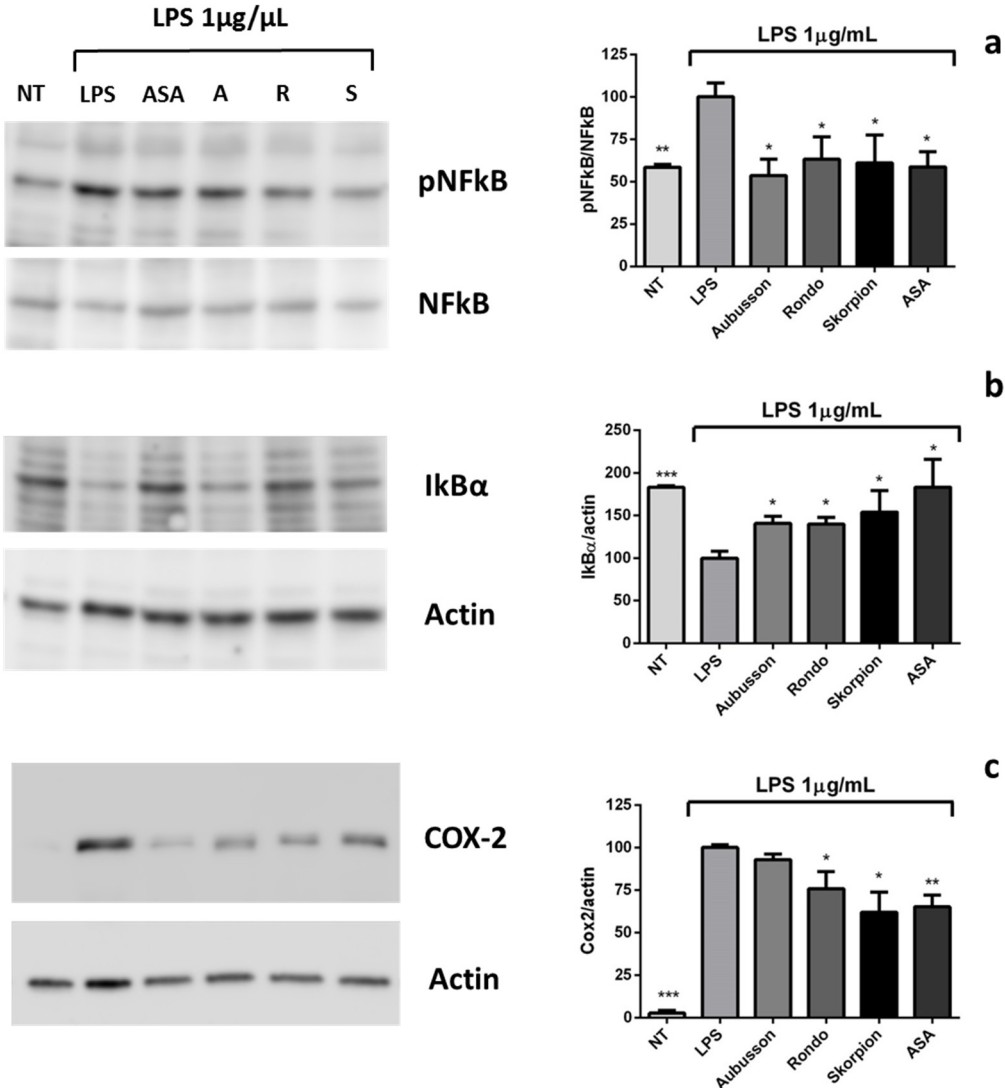

**Figure 5.** Western blot analysis (on the left) and relative protein levels (on the right) from cells treated with LMW peptides from breads. RAW 264.7 cells were incubated with LMW peptides (0.05 mg/mL) from the breads for 1 h and then with LPS for 24 h. (**a**) The intensity of the signal from anti-phospho-NFkB mAbs was normalized on the anti-total NFkB mAbs signal. (**b**) anti-IkB and (**c**) anti-COX2 mAbs signals were normalized on the actin signal. NT: no treated cells; LPS: cells treated with lipopolysaccharide; ASA: cells treated with LPS and acetyl salicylic acid. Aubusson, Rondo, and Skorpion indicate cells treated with both LPS and LMW peptides. Data reported in the figure represent the mean (±SD) of three different experiments performed in duplicate. Statistical analysis was performed for each sample vs. LPS. (* $p < 0.05$; ** $p < 0.01$; *** $p < 0.001$).

## 4. Discussion

Bioactive peptides from fermented foods have recently attracted attention for their antioxidant properties [31,32]. Peptides are fragments nascent in the primary protein sequences and could impart health benefits beyond basic nutritional advantages. It has been suggested that the amino acid composition can determine the radical scavenging, inhibition of lipid peroxidation, and metal ion chelation activity of peptides [33]. However, antioxidant and anti-inflammatory activities are interdependent since an oxidant stimulus can induce inflammation. Inflammatory cells liberate a number of reactive species at the site of inflammation, leading to oxidative stress; on the other hand, a number of reactive oxygen species can induce an intracellular signaling cascade that enhances the expression of proinflammatory genes [34].

Therefore, in order to define a food-derived molecule as a nutraceutical, it should at least have antioxidant and anti-inflammatory properties. In this respect, bioactive peptides are good candidates, since they often show the ability to regulate several inflammatory markers, such as inducible nitric oxide synthase (iNOS), cyclooxygenase 2 (COX-2), cytokines, and chemokines, that act in concert with NF-kB activation and IkB degradation on the inflammation pathway [23,35,36].

Healthy breads are defined as baked products that can have beneficial effects on human wellness, and sourdough fermentation greatly increases the sensory properties, shelf-life, and nutritional value of the products, and it naturally induces the production of bioactive peptides equipped with antioxidant and anti-inflammatory properties [1]. In particular, some LAB strains have been shown to produce doughs with greater functional and nutritional value than others and bioactive peptides derived from the doughs have been characterized [8,10,37]. Most of the peptides produced during fermentation are derived from the main proteins of the flour [10]. Therefore, it is of peculiar importance to select flours as well as LAB and yeast strains to obtain breads with high contents of functional compounds, such as polyphenols, free amino acids and derivatives, and bioactive peptides [5,37–39].

Previous studies have been performed on sourdoughs obtained with whole grain flour, including a hull-less barley with high β-glucan content, a blue-grain wheat flour (cv Skorpion) containing high anthocyanin concentrations, and a conventional red-grained wheat variety (cv Aubusson). They showed that barley sourdough possessed a strong antioxidant activity, generally higher than Skorpion samples, that, however, was greater than that recovered in Aubusson doughs [5]. Starting from these observations, these flours were used for the preparation of sourdoughs and breads.

Three types of sourdoughs, one for each type of flour, were obtained by using selected LABs and yeast, previously characterized for different technological and nutritional features [5,17,18]. The sourdoughs were then used to produce the respective breads. Soluble water extracts were obtained either from the sourdoughs and breads and used to fractionate the peptide content by reverse-phase chromatography. Low-molecular-weight (LMW) peptides, both from sourdoughs and breads, were easily recovered during elution and characterized for their functional properties by ex vivo assay on cultured murine macrophages. As expected, based on previous results [8], none of the LMW peptides had any cytotoxic effect even after 24 h of incubation with cultured cells. The mixtures of LMW peptides from each sample, both sourdoughs and breads, were then tested for the ability to reduce LPS-induced ROS production in macrophage cells. As observed for peptides derived from sourdoughs produced with commercial flour [10], all the peptides showed the ability to reverse the formation of free radicals induced by LPS itself. The cooking procedure did not change the antioxidant properties of peptides, which is in the same order as that obtained for cells treated with ascorbic acid. Moreover, while water-soluble extracts from sourdoughs obtained with hull-less barley showed an antioxidant activity stronger than that obtained with Skorpion and Aubusson extracts [5], no significant differences in ROS scavenging were observed for fractionated peptides derived from different flours. The result could be explained by the use of the same amount of peptides employed in the experiment, as well as by the presence of antioxidant molecules that are in the extract but that are lost during the isolation of peptides.

Since information about peptides from cooked breads might be useful for manufacturing new baked products with nutraceutical properties, further characterization of biological activity of peptides was performed only for LMW peptides obtained from cooked breads. Firstly, there is the ability to counteract the inflammation pathway induced by LPS. Relying on LPS' ability to induce the activation of inflammatory pathways, to increase the synthesis of reactive oxygen species, and consequently, to reduce cell viability [27], the protective effect of peptides from breads to recover cell vitality upon LPS treatment was tested. The peptides from Aubusson bread protect macrophages so much that the live cell number was the same as that obtained with untreated cells; in addition, the peptides from Skorpion and

Rondo breads even seem to have a positive effect on cellular health. The results from the anti-inflammatory activity show in the same trend: peptides from Skorpion and Rondo breads reduced, in significant manner, the expression of COX2, which was chosen as a downstream marker of the inflammatory pathway. On the contrary, the expression of both the pNFkB transcription factor and its inhibitor IkB show that all the peptides, irrespective of their origin, have the same anti-inflammatory activity.

On the whole, these results do not allow identification of the "best dough" to be used for the production of nutraceutical breads but indicate that the combination of flours and LAB and yeasts used in the present manuscript led to the production of bakery products with beneficial effects on cell viability, as well as on the oxidative and inflammatory status. Since these processes are simultaneously found in many chronic pathological conditions, constant consumption of beneficial foods could help in maintaining low inflammation and oxidative levels. Obviously, further analysis is needed to unravel the amino acid composition and sequence of peptides derived from these fermented breads in order to highlight the presence of sequence tags to be compared with peptides from other fermented foods. However, the present data have increased the knowledge about the beneficial effect of bioactive peptides from doughs that are retained in cooked breads and that could result in the preparation of new baked products from peculiar flours to be studies for their nutraceutical properties.

**Author Contributions:** Conceptualization, S.L.; methodology, S.L. and V.G.; validation, S.L. and M.V.; investigation, S.L.; resources, L.P.; data curation, V.G. and M.V.; writing—original draft preparation, L.P. and L.G.; writing—review and editing, P.P.; supervision, L.P. All authors have read and agreed to the published version of the manuscript.

**Funding:** Research funded by Ministero dell'Istruzione, dell'Università e della Ricerca (PRIN2015SSEKFL), Italy.

**Informed Consent Statement:** Not applicable.

**Data Availability Statement:** Not applicable.

**Conflicts of Interest:** The authors declare no conflict of interest.

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
