# Peer review of "Bioactive Properties of Breads Made with Sourdough of Hull-Less Barley or Conventional and Pigmented Wheat Flours"

_applsci, doi:10.3390/app11073291_

Round 1

Reviewer 1 Report

Dear Authors,

The manuscript is written at a very high professional level, therefore, I have found some minor editorial rather than methodical errors. The ms is adequately divided into individual chapters, the abstract is concise, the literature background in introduction is focused on the topic, the methodology is described accordingly, the data are properly statistically processed, clearly presented, and adequately commented in the discussion. The conclusions of the study were described correctly and are based on the results obtained earlier.

Materials and methods section: A description of the statistical methods used are missing.

The manuscript is well written and its contents are appropriate but I noticed some editorial errors e.g. punctuation marks. Other minor flaws:

Line 64 - Furfurilactobacillus . rossiae  - there is dot between genus and species

Line 97 – typo “the”

Line 114 and 116 – Latin name of Lp. plantarum should be written: L. plantarum

Line 159 – in this case the names of authors should be placed before citing them numerically

Line 205-206 – At the end of the sentence there is lack of the method reported to citation nb 22. So it should be: …. reported by Galli et al. [22].

Line 227: (25° C) – check the guidelines for authors – sometimes you are using space between sign of Celsius and the value and sometimes not – e.g. 25 °C or  25° C.

Line 228 – there is extra space between the sentences

Line 279 – wrong table number – it should be Table 3

Line 302, 327 - ( ± SD) – there is extra space between ( and plusminus sign

At the end I would like to congratulate the Authors of excellent study, my recommendation is to accept paper with minor revision.

Author Response

REVIEWER #1

Comments and Suggestions for Authors

Dear Authors,

The manuscript is written at a very high professional level, therefore, I have found some minor editorial rather than methodical errors. The ms is adequately divided into individual chapters, the abstract is concise, the literature background in introduction is focused on the topic, the methodology is described accordingly, the data are properly statistically processed, clearly presented, and adequately commented in the discussion. The conclusions of the study were described correctly and are based on the results obtained earlier.

Materials and methods section: A description of the statistical methods used are missing.

 Thank you!, I really forgot to insert them. I added the 2.7 subsection for Statistical analysis

The manuscript is well written and its contents are appropriate but I noticed some editorial errors e.g. punctuation marks. Other minor flaws:

Line 64 - Furfurilactobacillus . rossiae  - there is dot between genus and species.

That’s right! I took it of

Line 97 – typo “the”

It has been added

Line 114 and 116 – Latin name of Lp. plantarum should be written: L. plantarum

I changed the name as you indicated.

Line 159 – in this case the names of authors should be placed before citing them numerically

The name of the first author has been added.

Line 205-206 – At the end of the sentence there is lack of the method reported to citation nb 22. So it should be: …. reported by Galli et al. [22].

I inserted the name as you suggested.

Line 227: (25° C) – check the guidelines for authors – sometimes you are using space between sign of Celsius and the value and sometimes not – e.g. 25 °C or  25° C.

By checking the guidelines for authors, I saw that the correct form is 25 °C; therefore, I replaced thi form in the text

Line 228 – there is extra space between the sentences

I deleted it.

Line 279 – wrong table number – it should be Table 3

Thank you for the observation, I did not notice that

Line 302, 327 - ( ± SD) – there is extra space between ( and plusminus sign

 I corrected all the legends in which the ( ± SD) was misspelled

At the end I would like to congratulate the Authors of excellent study, my recommendation is to accept paper with minor revision.

Thank you for the comment. I hope to be able to continue working on these topics with ever better results

Reviewer 2 Report

The manuscript reported by Luti et al., reports a potential application of bioactive Low Molecular Weight (LMW) peptides obtained from breads and sourdoughs with selective LAB in food production. The manuscript was well written and the results are interesting and potential for application. The manuscript can be improved if the authors address some questions as below:

  1. Although the authors wrote in “Discussion” (line #460-line #463) that it is essential to identify amino acid composition and sequences of peptides in “LMW peptides” in the future. However, to make sure that the LMW peptides used in this study are bioactive peptides, the authors still need to provide information of LMW peptides they used. What are they? How to know they are bioactive peptides just based on their definition that they collected the “5-20 min” fraction from result in Figure 1? It is important to provide the second evidence to prove that the “the 5-20 min fraction” in this study actually contained LMW peptides.
  2. Why did they use 0.05 mg/ mL LMW peptides in their experiments in Figure 2, 3, 4 and 5? Why did the author treat the cells for 1 h with LMW peptides? The authors should perform treatment with different concentration of LMW peptides and time-course treatment of LMW peptides.
  3. In Figure 5, it is interesting to see the reduction of COX-2 expression after treating cell with LMW peptides from beads. However, activity of COX-2 is essential to analyze in this experiment.

Author Response

REVIEWER #2

Comments and Suggestions for Authors

The manuscript reported by Luti et al., reports a potential application of bioactive Low Molecular Weight (LMW) peptides obtained from breads and sourdoughs with selective LAB in food production. The manuscript was well written and the results are interesting and potential for application. The manuscript can be improved if the authors address some questions as below:

  1. Although the authors wrote in “Discussion” (line #460-line #463) that it is essential to identify amino acid composition and sequences of peptides in “LMW peptides” in the future. However, to make sure that the LMW peptides used in this study are bioactive peptides, the authors still need to provide information of LMW peptides they used. What are they? How to know they are bioactive peptides just based on their definition that they collected the “5-20 min” fraction from result in Figure 1? It is important to provide the second evidence to prove that the “the 5-20 min fraction” in this study actually contained LMW peptides.

Thank you for the observation. We are usually do electrophoretic analysis on each sample before and after the HPLC fractionation either in this and in previous works.  I really forgot to insert the experiment in results and I’m adding the respective figure in the 3.2 subsection.

  1. Why did they use 0.05 mg/ mL LMW peptides in their experiments in Figure 2, 3, 4 and 5? Why did the author treat the cells for 1 h with LMW peptides? The authors should perform treatment with different concentration of LMW peptides and time-course treatment of LMW peptides.

The concentration of peptides used in experiments as well as the 1 h treatment have been set up during the preparation of previous works (Galli et al.,  Int. J. Food Microbiol 2018, 286, 55–65, doi:10.1016/j.ijfoodmicro.2018.07.018; Luti et al., Food Chemistry 2020, 322, 126710, doi:10.1016/j.foodchem.2020.126710). in which, on the basis of literature data, we tried different incubation conditions: (i) the co-incubation treatment; (ii) the pretreatment with peptides for 1 h followed by LPS treatment for 24 h, ; (iii) a pre-treatment with LPS, that is, however, not utilized in many such studies as confirmed by several manuscripts, among which (just to name a few): Cheng et al., Molecules 2017, 22(7), 213;  https://doi.org/10.3390/molecules22071213; Hyo-Jung et al.,  Mol. Immunol. 2020, 119; 123-131 https://doi.org/10.1016/j.molimm.2020.01.010; Asanka Sanjeew el al.,  Biomolecules 2020, 10(4), 511; https://doi.org/10.3390/biom10040511;  Cao et al., Nutrients 2019, 11(11), 2794; https://doi.org/10.3390/nu11112794.

Since pre-treatment with peptides had given us the best results, we used the same condition also in this work. The time of incubation with peptides (1 h) and the following LPS treatment (24 h) have been established on the literature data, too. Finally, final concentration of peptides (0.05mg/mL) was in the same range as the one used in our previous works. The right concentration was previously determined by several tests with 0.005-0.5 mg/mL peptides.

A sentence about the used condition in treatment of cells has been added in the revised version in the 2.6.3 subsection.

  1. In Figure 5, it is interesting to see the reduction of COX-2 expression after treating cell with LMW peptides from beads. However, activity of COX-2 is essential to analyze in this experiment.

I totally agree with you that determining enzymatic activity is the best way to know the role of an enzyme and Cyclooxygenase (COX) Activity Assay Kit (Fluorometric) (ab204699) can detect activity of both total and specific isoforms by the use of specific inhibitors. The method has been used by some researchers (for example by . De stefanis et al., Investigating the Connection Between Endogenous Heme Accumulation and COX2 Activity in Cancer Cells. Front Oncol. 2019 Mar 19;9:162. doi: 10.3389/fonc.2019.00162), but it is not much used. Even when the word “activity” is cited in the title , the manuscripts often report the expression analysis performed by specific antibodies and/or PCR primers. Alternatively, the assay is made by detecting the amount of products such as PGE2 and TXB2 (Dec et al. Long-term exposure to fluoride as a factor promoting changes in the expression and activity of cyclooxygenases (COX1 and COX2) in various rat brain structures. Neurotoxicology. 2019 Sep;74:81-90. doi: 10.1016/j.neuro.2019.06.001) .

However,  the most widely used method to detect the role of COX2 in the inflammatory pathway is the analysis of the expression profile by the use of specific antibodies. The method offers a good  indication of the inflammation process since COX2 in not expressed in normal tissue and its expression goes along with the induction of inflammation.in fact, it is well known that an increased expression of COX2 is strictly related with enhanced inflammatory response whereas a decrease o COX expression suggests a lower activity. Finally, determination of expression level of COX2 and other inflammatory markers is currently used in experiments dealing with the beneficial role of bioactive peptides from foods as widely reported in literature: Li et al Potential mechanisms underlying the protective effects of Tricholoma matsutake singer peptides against LPS-induced inflammation in RAW264.7 macrophages.   Food Chem. 2021 Mar 3;353:129452. doi: 10.1016/j.foodchem.2021.129452; Yi et al; Soybean protein-derived peptides inhibit inflammation in LPS-induced RAW264.7 macrophages via the suppression of TLR4-mediated MAPK-JNK and NF-kappa B activation. J. Food Biochem. 2020 Aug;44(8):e13289. doi: 10.1111/jfbc.13289.

Therefore we chose the total amount of COX2 instead of its activity to highlight the anti-inflammatory properties of peptides form breads. A sentence dealing with the chosen method has been added in 3.5 section.

Round 2

Reviewer 2 Report

Although the author cannot address my third question from my last review comment, but their response sounds reasonable. There is only one thing I would like the authors to confirm: as they wrote in line #224-225 “The numerical results of microbial and chemical analysis in this study are averages of two or three independent replicates”. If any experiment they have done only two independent replicates, I suggest the authors include at least one more replicate to confirm their result.

Author Response

Dear Reviewer, 

regarding your observation, I think you refer to 244-245 lines of the revised version. You are right, the sentences is written ambiguously. Determination of acidification (pH, total titratable acidity), volume increase (%, ∆V/V0) and plate counts of LABs and yeasts were performed in triplicate. No experiment has been done with only two replicate, the word “two” reported in line 244 was probably due to the number of plates counted for bacteria and yeasts.  Really, as reported in line # 150, regarding plate counts, we meant two plates for each dilutions, however in order to avoid any misunderstanding we modified the text. 

To confirm the goodness of the method, I can underline that it has been used several times (for example in Ref#19:  Galli et al., Liquid and firm sourdough fermentation: microbial robustness and interactions during consecutive backsloppings. LWT 2019, 105, 9–15, doi:10.1016/j.lwt.2019.02.004.  and in Ref #5, Galli et al.; Antioxidant Properties of Sourdoughs Made with Whole Grain Flours of Hull-Less Barley or Conventional and Pigmented Wheat and by Selected Lactobacilli Strains. Foods 2020, 9, 640, doi:10.3390/foods9050640); moreover,  also in this work microorganism concentrations, for both lactic acid bacteria and yeasts in sourdough bread were in the typical range of sourdough products as reported in the text (line 271-272).

I hope I have clarified the problem, probably due to subsequent corrections by co-authors and not reviewed carefully enough by myself